# Novel Physical and Biological Applications of Carbon Ion Radiotherapy

**DOI:** 10.3390/cancers18010113

**Published:** 2025-12-30

**Authors:** Danushka Seneviratne, Prapannajeet Biswal, Sunil Krishnan

**Affiliations:** 1Department of Radiation Oncology, The University of Oklahoma College of Medicine, Oklahoma City, OK 73104, USA; 2Vivian L. Smith Department of Neurosurgery, UTHealth Houston, Houston, TX 77030, USA; prapannajeet.biswal@uth.tmc.edu (P.B.); sunil.krishnan@uth.tmc.edu (S.K.)

**Keywords:** carbon ion radiotherapy, linear energy transfer, DNA repair, immunomodulation, hypoxia

## Abstract

A growing body of evidence suggests that, much like proton beams that are currently available at multiple centers across the United States, heavy ions (including carbon ions) can also be directed precisely to tumors with minimal dose beyond the Bragg peak and tight lateral penumbras. However, compared to proton beams, carbon ion beams have higher linear energy transfers and relative biological effectiveness, though they do possess a short disintegration tail past the Bragg peak. Here, we highlight key attributes of carbon ion beams that can be exploited for therapeutic gain, namely, complex DNA damage that is harder to repair, immune activation, and less susceptibility to hypoxia-triggered radiation resistance. We also highlight key limitations of carbon ion therapy that preclude widespread adoption of this technology in clinical practice.

## 1. Introduction

Particle therapy in radiation oncology may be roughly categorized into either “proton therapy” or “heavy ion therapy” based on the physical and biological properties of such treatment. The term “heavy ions” indicates that the particle employed weighs more than a ^4^He atoms [1]. As these charged particles approach the end of their range, they slow down significantly and deposit large amounts of energy over a short distance, creating the characteristic Bragg peak. By scanning this Bragg peak across a tumor, high doses of radiation can be selectively deposited in the tumor, while largely sparing adjacent tissues. Proton therapy exploits this Bragg peak effect and limits dose to tissues distal to the tumor, but suffers from a high lateral scatter and therefore, a larger lateral penumbra. The relative biological effectiveness or RBE (the ratio of biological effectiveness of one type of ionizing radiation relative to standard irradiation with cobalt gamma rays when given the same amount of absorbed energy) is generally assumed to be constant at 1.1; i.e., a slightly improved (~10%) biological effectiveness over conventional X-ray irradiation [2]. In passive scattering mode, where a large area is irradiated at one time, multiple Bragg peaks of varying penetration depths and variable intensities are stacked together to produce a spread-out Bragg peak (SOBP) that covers the entire tumor. The RBE is empirically assigned to be 1.1 within this SOBP. An alternative proton beam delivery method uses small pencil beams of protons raster-scanned across the tumor target to dose-paint the tumor with the desired dose. These beams are magnetically steered in the x-y plane, and energy selection drives the steering in the z plane. Carbon ion radiation therapy (CIRT) also delivers precise radiation therapy to a tumor while sparing surrounding normal tissue, using beams of accelerated carbon ions rather than protons. CIRT is distinguished from commonly used X-ray (or photon) and proton radiation by its high density of ionization and the large amount of dose deposited per unit length traversed (linear energy transfer, LET). The consequent increase in biological damage, as discussed further below, leads to an RBE that ranges from 3 to 4 and is variable along the SOBP [2]. For instance, the LET of proton beams in the entrance region is 1 keV/µm, similar to that of clinical megavoltage photon beams, and rises to about 10 keV/µm at the Bragg peak, whereas the LET of carbon ion beams can be as high as 80–100 keV/µm at the Bragg peak. Furthermore, owing to the larger mass of carbon ions, multiple Coulomb scattering of CIRT compared to proton therapy is markedly reduced, leading to a sharply delineated lateral penumbra [3]. It also differs from proton radiation, which uses much lighter accelerated charged ions, and because of the significantly greater mass of carbon ions, the particle accelerators (synchrotrons), gantries, and beam shaping assemblies needed to generate, aim, and steer carbon ion beams across a tumor with pinpoint precision are considerably larger, heavier, more expensive, and more complex. Only a handful of centers globally can offer this form of radiation therapy to patients owing to the structural and engineering challenges. Ongoing clinical trials at these centers are continuously seeking to define the therapeutic benefits, if any, of CIRT over traditional radiation therapy techniques [4,5,6].

Although CIRT has several unique and advantageous properties, such as its sharp lateral penumbra and steep lateral dose fall-off beyond the target volume, much remains to be understood before we can fully exploit its potential to advance cancer care. In terms of its biological impacts, CIRT must be contrasted with conventional low-LET X-ray and proton-based therapies, which primarily exert their anti-tumor effects through indirect mechanisms where ionization of molecules such as water generates oxygen-free radicals that trigger DNA damage. In contrast, heavily charged particles such as carbon ions have a high density of ionization and a high LET, resulting in more pronounced direct DNA damage, where the ionization of nucleotides in the DNA directly triggers changes in DNA strands, nucleotides, and bases. Direct DNA damage induced by CIRT is characterized not only by more DNA damage but also by more complex DNA damage. A key feature of this complex DNA damage is clustering of multiple individual lesions, such as DNA double-strand breaks, DNA single-strand breaks, and base damage, within one or two helical turns of DNA. As a result, the RBE of high-LET radiation is higher than that of low-LET radiation (such as X-rays) because these clustered DNA lesions are less amenable to repair than scattered individual DNA lesions induced by low-LET radiation. This distinction lies at the heart of what makes CIRT different from other forms of radiation [5,7,8,9].

Another unique feature of CIRT is its relatively greater effectiveness in treating hypoxic tumors. Hypoxic tumors tend to be, on average, three times more resistant to radiation than normoxic tumors. This is expressed as an oxygen enhancement ratio (OER) of 3, where the dose needed to achieve the same biological effect (say, 10% survival on a classical colony formation assay) is three times higher for hypoxic cells/tumors than normoxic cells/tumors. Given the aforementioned dependence of CIRT-induced DNA damage on the direct effect, there is less need for oxygen-free radicals to damage DNA, and therefore, the biological effect is less dependent on the oxygenation status of the cell/tissue. Nonetheless, the OER for CIRT is not 1 (i.e., normoxic and hypoxic tumors respond equally well to CIRT), but experimental results suggest that it can be considerably less than 3 [10]. An added layer of complexity is posed by hypoxia being either acute or chronic, which of these is measured or imaged, and whether CIRT overcomes the detrimental effect of one or both or neither. However, much research is still needed to fully understand the biological mechanisms underlying the therapeutic benefits of CIRT and to optimize its use in clinical settings.

Inducing cancer remission by stimulating the body’s natural immunity (immunotherapy) has evolved to become a core component of the multimodal treatment of advanced malignancies, as demonstrated by data from multiple Phase III trials [11]. Radiotherapy has been known to enhance the effects of immune checkpoint inhibitors (ICIs) at the primary tumor site and also induce regression of distant, nonirradiated sites—termed the ‘abscopal effect’ [12,13]. Ideally, the radiation should be able to stimulate an immunologic boost by causing significant local damage but concurrently sparing the peritumoral tissues, the tumor-infiltrating lymphocytes, and the draining lymph nodes [14]. The property of carbon ions to deliver high-energy doses in the Bragg peak while sparing peritumoral damage makes for a promising choice of radiotherapy to enhance immunogenicity of solid tumors.

The purpose of this review is to explore how the unique biological properties of CIRT can be harnessed to develop novel applications. Specifically, we will discuss how CIRT can modulate the tumor immune response and increase tumor cell kill in traditional radiation-resistant hypoxic tumors by virtue of its high-LET nature.

## 2. Immunomodulatory Effects of Radiation

Over the last decade, clinical findings have triggered investigations into the impact of radiation on the immune system in order to exploit these interactions for improved cancer outcomes. A groundbreaking paper published in the *New England Journal of Medicine* in 2012 reported the observation of abscopal effects in a melanoma patient who received local irradiation and immunotherapy with ipilimumab [15]. Abscopal effects occur when systemic effects are observed after localized radiation therapy, and have been anecdotally noted for several decades. The publication of this discovery resulted in a rapid escalation of research into the immunological effects of radiation therapy. For example, a secondary analysis of the Keynote 001 trial, which was originally designed to evaluate the effects of PD-1 inhibition with pembrolizumab in patients with locally advanced or metastatic non-small cell lung cancer (NSCLC), demonstrated that outcomes were significantly improved in patients who received extracranial radiation therapy prior to immunotherapy compared to those who received immunotherapy alone [16]. Similarly, in the Pacific Trial, patients with locally advanced NSCLC who received immunotherapy following thoracic radiation had significantly improved survival rates [17]. At the molecular level, the interactions between radiation therapy and the immune system are highly complex and not fully understood. However, it is generally believed that macromolecular damage associated with radiation leads to the production of immunostimulatory epitopes [18,19]. The immunogenic cellular response to any stressor is elicited primarily by damage-associated molecular patterns (DAMPs), which are immune-stimulatory molecules constitutively expressed by dying cells. DAMPs include a variety of molecules, such as nucleic acids, mitochondrial DNA, endoplasmic reticulum resident proteins that translocate to the cell surface like calreticulin, heat shock proteins, cytokines such as Type 1 interferons, high-mobility group box-1 (HMGB1), and ATP, which collectively act as an “eat me” signal for antigen-presenting cells and dendritic cells [20].

The immunomodulatory effects of X-ray radiation are believed to begin following the induction of genotoxic stress, which, in combination with defective DNA repair, replication stress, and altered cell cycle progression (G2-M phase), can lead to the accumulation of fragmented DNA and micronuclei in the cytoplasm. The presence of cytosolic DNA then activates the cyclic guanosine monophosphate–adenosine monophosphate GMP-AMP synthase (cGAS) stimulator of interferon genes (STING) pathway. Following the binding of cGAS to cytosolic DNA, its enzymatic abilities are activated, causing the synthesis of the second messenger 2′,3′-cyclic GMP–AMP (cGAMP). STING is an endoplasmic reticulum protein that serves as a major immune modulator. The binding of cGAMP to STING causes a structural change and translocation of the STING complex to the Golgi apparatus. It then serves to recruit additional downstream proteins, including TANK-binding kinase signaling (TBK1), which phosphorylates STING, leading to activation of interferon regulatory factor 3 (IRF3) kinase. IRF3 moves to the nucleus and enables the transcription of Type I interferons, initiating dendritic cell maturation, migration of dendritic cells to regional lymph nodes, and the expansion of cytotoxic T-lymphocytes in regional lymph nodes. Eventually, this promotes immunogenic cancer cell death [21,22]. This mechanism of immune activation by cytosolic DNA micronuclei via the cGAS-STING pathway can be enhanced by the use of CIRT, as it is more efficient in the production of micronuclei compared with fractionated proton therapy over similar dose ranges [14,23]. The immunomodulatory potential of CIRT at a molecular level is demonstrated in Figure 1.

## 3. CIRT’s Potential as an Immunogenic Cancer Treatment

Although the immunomodulatory effects of X-rays have been extensively studied in vitro and in in vivo animal models, there is a paucity of preclinical and clinical data regarding the impact of CIRT on immune signaling [24]. The potential of CIRT to transform an “immunologically cold” tumor into a “immunologically hot” one is often discussed because CIRT causes massive molecular damage and triggers multiple non-traditional cell death pathways, thus potentially having greater immunomodulatory potential than X-rays [25,26]. For example, heavy ions can activate alternative cell death pathways independent of TP53 and Bcl2, which may increase the release of immunogenic neoantigens [25]. In nasopharyngeal carcinoma cell lines, MLKL, a central executor protein of necroptosis, was upregulated significantly (up to 28-fold increase) following CIRT compared to photon irradiation at equivalent physical (4 Gy) and RBE (10 Gy) doses [27]. Following carbon ion exposure, human glioblastoma cell lines and cervical cancer cell lines showed a significant unfolded protein response, an increase in LC3-II (as indicated by the presence of increased acidic vesicular organelles or AVOs with 24.9% of cells irradiated with CIRT compared to just 8.68% exposed to X-rays) and a decrease in p62 protein levels (just 22% expression in SHG44 cell lines and 16% expression in HeLa cell lines on western blotting compared to 100% after X-ray irradiation) indicating that autophagy was the primary mechanism of cell death in response to CIRT [28]. CIRT induces autophagy in a dose-dependent and LET-dependent manner by suppressing the PI3K-Akt pathway [29,30,31,32]. The ability of CIRT to elicit highly immunogenic forms of cell death, such as autophagy and necroptosis, further supports its immunostimulatory potential, which could be utilized to improve tumor control.

The transfer of calreticulin to the surface of tumor cells has been observed following X-ray and proton radiation, which increases their sensitivity to T-lymphocytes. However, studies have shown that this effect is significantly stronger in response to irradiation with carbon ions [33]. This suggests that the differential nature of macromolecular damage induced by CIRT is able to elicit DAMPs to a greater extent than traditional radiation modalities [25,33]. The heightened production of DAMPs and varied epitopes can lead to increased uptake of tumor antigens by dendritic cells, promoting T cell differentiation into CD8+ cytotoxic T lymphocytes and activating anti-tumor immunity [26,34]. In addition, CIRT effectively triggers canonical signs of immunogenic cell death (ICD), including calreticulin translocation and HMGB1 and ATP release. Calreticulin and HMGB1 induce dendritic cell maturation to enhance tumor antigen presentation to T cells, and mature dendritic cells express cytokines, chemokines, and costimulatory molecules (CD40, CD80, and CD86) [35]. Ken Ando et al. have demonstrated that a CIRT-dendritic cell combination immunotherapy suppresses pulmonary metastasis in murine models [36]. Recent studies in sarcoma mouse models comparing photons to carbon ions have also shown that combining CIRT with anti-PD-1 therapy results in greater infiltration of CD4+ and CD8+ lymphocytes (~ 20% more CD4+ cells and 25% more CD8+ cells were detected per field-of-view on immunohistochemistry) into the tumor bed and decreased tumor growth [37].

Given the immense potential of the cGAS-STING pathway for triggering immune signaling and priming T cells, there has been increasing interest in evaluating the impact of CIRT on the activation of this pathway. Studies to date have shown that CIRT can lead to increased micronuclei production in comparison to X-rays [38,39]. For instance, in Chinese hamster V79 cells exposed to carbon ions and Cobalt-60 gamma rays, the high-LET carbon ions resulted in significant increases in micronuclei production and DNA ladder formation [40]. As the accumulation of cytosolic DNA serves as the initiating event for activating the cGAS-STING pathway, the increased micronuclei formation following CIRT may significantly enhance cGAS-STING signaling. In fact, cell studies have demonstrated that cGAS activation is dependent on dose, increasing proportionally to the amount of cellular double-stranded DNA present [34]. While this concept requires further investigation, recent preliminary data in melanoma mouse models indicate that CIRT leads to increased cGAS-STING pathway activation and is associated with greater tumor regression compared to X-rays. Additionally, CIRT has been found to enhance tumor infiltration of natural killer cells, CD4+ and CD8+ T lymphocytes, and gene expression of PD-1, lymphocyte activation gene, and T cell immunoglobulin. Preclinical studies have reported enhanced PD-L1 expression on tumor cells after CIRT. Permata et al. noted that PD-L1 upregulation is predominantly driven by Rad3-related (ATR) kinase activation and subsequent ATR-Chk1 axis activation secondary to single-stranded DNA breaks (SSBs) and double-stranded DNA breaks (DSBs), which was diminished by ATR kinase inhibition. In this study, CIRT induced greater PD-L1 expression compared to photon RT [41]. PD-L1 increase after CIRT seems to be cell line agnostic. In human cervical squamous carcinoma cells, a dose-dependent increase in PD-L1 mRNA and protein levels was recorded, attributable to increased phospho-Chk1 (p-Chk1). On inhibition of p-Chk1, the PD-L1 upregulation was abrogated [42]. Defects in non-homologous end joining and homologous recombination both augment DSBs, which can drive PD-L1 expression. However, STAT1/1-IRF1 pathway activation seems indispensable for this process to occur. Since high-LET CIRT induces more DSBs (more numerous and complex breaks), a greater activation of such signaling pathways downstream of DSBs may drive PD-L1 expression on tumor cells [43].

Combining PD-L1 inhibitors with CIRT increases natural killer and T cell infiltration within the tumor microenvironment and reduces tumor growth compared to individual therapies [44]. Overall, these studies suggest that CIRT may act as a “cancer vaccine” that can better mobilize the immune system, potentially leading to improved patient outcomes, particularly in conjunction with immunotherapy [25]. Further investigations of the immunogenic potential of CIRT in various cell types and animal models must be conducted prior to assessing this concept in human clinical trials. Ultimately, transforming immunologically “cold” tumors through CIRT could significantly improve patient outcomes in radiation-resistant malignancies. The following table (Table 1) provides brief insights into the various novel mechanisms of immune stimulation by CIRT.

The major mechanisms of CIRT-induced immunogenic cell death observed across diverse cell lines may be distilled as follows:(1)CIRT triggers “eat-me” signals (calreticulin translocation to the cell membrane and secretion of HMGB1 and ATP) more than low-LET RT and activates novel cell death mechanisms such as ferroptosis and necroptosis, all of which trigger the release of DAMPs.(2)CIRT-induced complex dsDNA breaks and micronuclei trigger cGAS-STING activation and type-1 IFN response, leading to innate immune cell activation and infiltration (macrophages, dendritic cells and NK cells). This process is complemented by the DAMP release.(3)Autophagy counteracts the above processes, possibly by accelerating the cytoplasmic micronuclei cleanup, thereby preventing cGAS-STING activation. Autophagy inhibitors may therefore be used to increase the potency of CIRT even further. TREX1 also senses and degrades dsDNA to dampen cGAS-STING activation.(4)The pro-inflammatory tumor microenvironment (resulting from #1 & 2) triggers CD8+ infiltration into tumors, lymph node or splenic T cell proliferation, and memory T cell generation, which drive the abscopal anti-tumor response. This is complemented by CIRT reducing Treg or MDSC proliferation as opposed to X-Ray RT.(5)CIRT induces PD-L1 expression more than X-ray RT owing to its ability to generate complex nuclear damage. Unsurprisingly, this process is exacerbated by defective DNA repair pathway proteins (ATM/ATR/BRCA). This opens the doors for anti-PD-1/PD-L1 combination immunotherapy.

In addition to its potential for local tumor control, preliminary data with particle therapy show promise in reducing the metastatic potential of tumors. For example, in a study by Yang and colleagues, culture medium from A549 lung cancer cells exposed to CIRT effectively inhibited the metastasis of tumor cells, as both migration and invasion rate reduced by 20% [70]. Additionally, Akino and colleagues found that metastasis-associated gene expression patterns were downregulated in NSCLC cell lines following CIRT [71]. In particular, mRNA expression of ANLN, a homologue of anillin, was suppressed to about 60% of the basal level of expression 12 h after CIRT. Treatment with both proton and carbon ions in fibrosarcoma cell lines caused a greater dose-dependent decrease in cell migration and invasion compared to X-rays. In osteosarcoma mouse models, carbon ion irradiation resulted in a dose-dependent decrease in pulmonary metastases formation compared to X-rays, leading to a 20% decrease in the number of metastatic lung nodules compared to the control group following 1 Gy RBE of CIRT and a 60% decrease following 5 Gy RBE of CIRT [72]. If confirmed in further studies, this potential to reduce metastases could justify the use of CIRT in malignancies such as lung and pancreatic cancer, where metastatic spread remains the primary driver of mortality.

The immunogenic and anti-metastatic potential of CIRT suggests an increased potential for exploring abscopal effects compared to traditional radiation modalities. Although there is no level I evidence, published clinical observations suggest that CIRT may have enhanced abscopal capabilities, particularly in the treatment of bulky tumors. The National Institute of Radiological Sciences group reported anecdotal evidence of abscopal responses in two recurrent colorectal cancer patients treated with CIRT [73]. Tubin and colleagues have published on the concept of partial carbon ion irradiation of bulky tumors leading to abscopal effects. Among the 11 patients treated with CIRT without any systemic therapies, an average tumor volume regression of 61% was observed, and approximately 60% of the patients were noted to have abscopal effects at untreated locations [74]. While these findings are hypothesis-generating, further studies are necessary to assess the validity and therapeutic potential of this concept. Table 2 lists the NCT-registered clinical trials combining immunotherapy with CIRT.

## 4. The Potential for CIRT to Overcome Hypoxia-Induced Radiation Resistance

Recent studies have suggested that radiation resistance is a contributing factor to poor oncologic outcomes in subsets of patients with various malignancies, including head and neck cancers, cervical cancer, pancreatic adenocarcinoma, gliomas, adenocarcinomas of the gastrointestinal tract, prostate cancer, and NSCLC. A multitude of factors are thought to be implicated in radioresistance, such as tumor genomics, growth factor signaling pathway activation, the presence of cancer stem cells, and the presence of hypoxia. Hypoxia, in particular, has long been identified as an independent adverse prognostic factor and a strong contributor to tumor radioresistance. Unlike traditional radiation modalities that cause indirect DNA damage through the generation of oxygen-mediated free radicals, given that CIRT causes direct and clustered DNA damage in a less oxygen-dependent manner, it is thought to be advantageous in the treatment of hypoxic tumors.

The oxygen enhancement ratio (OER) is defined as the ratio of doses required to elicit the same biological effect under both normoxic and hypoxic conditions. For X-rays, the OER is often quoted as being approximately 3. Many studies have highlighted the low OER of carbon ions and the decrease in OER along the spread-out Bragg Peak in comparison to X-rays. Given this, CIRT is considered a viable option in the treatment of radioresistant, hypoxic malignancies. This concept has been demonstrated in several preclinical and clinical models. For instance, in a study involving NSCLC cell lines, Klein and colleagues found that CIRT had four times greater impact on cell kill than X-rays in both oxygenated and hypoxic conditions, and no significant oxygen effects were noted with the use of carbon ions [75]. In rat prostate carcinoma xenografts, Glowa et al. reported that compared to photons, CIRT required a lower (up to 17%) dose increase to result in equivalent tumor growth control in hypoxic conditions compared to normoxic conditions [76]. Further, in murine colon carcinoma cell lines treated under hypoxic conditions, CIRT resulted in greater cell growth inhibition compared to photon and proton RT at various doses ranging from 0.1 to 10 Gy [77]. While CIRT outperforms photon RT under hypoxic conditions, Valable et al. noted that erythropoietin knockout (KO) in glioblastoma cells still blunts the effectiveness of CIRT under hypoxia [78]. In another study involving various glioblastoma cell lines, the authors found that carbon ions led to more effective cell kill than X-rays under both normoxic and hypoxic conditions, although the extent of this impact was cell line-dependent [78]. A prospective Phase I/II trial involving CIRT and gemcitabine chemotherapy involving 72 patients with locally advanced pancreatic cancer (LAPC) found the overall survival rate (73%) and CT-based local control rate (83%) to be significantly higher than reported by Combs et al., with overall survival of 36% and localized progression-free survival at 13% [79,80]. In head and neck squamous carcinoma cell (HNSCC) lines, cells were more resistant to X-ray irradiation under hypoxia compared to normoxia, but this observed OER was no longer present when the cell lines were irradiated with carbon ions. Additionally, the authors found that HIF-1α was readily expressed in cancer stem cells following exposure to hypoxia, and the production of large amounts of reactive oxygen species (ROS) following photon radiation stabilizes HIF-1α, contributing to the radiation resistance observed following X-ray irradiation. In contrast, the limited ROS production following CIRT largely prohibits HIF-1α stabilization and limits the potential for hypoxia-induced radiation resistance [81].

In a separate study, the same authors performed studies in HNSCC cancer stem and non-stem cells to better understand the relationships between DNA damage, hypoxic tumor microenvironment, and DNA repair pathways following X-ray vs. CIRT. The detection and repair of double-strand breaks were lower in the carbon ion cohort compared to its X-ray counterpart. Interestingly, when photon irradiation was performed under hypoxia, there was accelerated initiation of both homologous and non-homologous repair pathways in both cancer stem cells and non-cancer stem cells, but this was not observed following CIRT. The authors suspected that this increased DNA repair signaling was likely contributing to hypoxia-induced radiation resistance following photon therapy. Overall, these studies suggested that carbon ions may be able to overcome radiation resistance in HNSCC by limiting the induction of DNA repair signaling in both cancer stem and non-stem cells [82]. Tinganelli and colleagues recently benchmarked the radiosensitivity of mammalian cells irradiated with carbon (and other) ions across a range of normoxic (21% oxygen concentration) to anoxic (0%) conditions [83]. They then developed a model that linked oxygen concentration and particle energy with OER, which was subsequently integrated into clinical treatment planning systems. These treatment plans were customized to ensure uniform cellular cytotoxicity throughout the treatment volumes with varying levels of radiosensitivity dictated by oxygenation status. Finally, they validated these adaptive treatment plans in two accelerator facilities using biological phantoms that had differing levels of oxygenation.

Despite the promising preclinical data described above, our understanding of the mechanistic underpinnings and the best clinical applications of CIRT-mediated hypoxic radiosensitization remains limited. Therefore, before we can use CIRT to treat hypoxic, radioresistant malignancies, we need to develop reliable methods for identifying clinically relevant hypoxic tumor subsets, determine whether it is safe and effective to deliver adaptive radiation therapy with CIRT specifically to hypoxic tumor regions, better understand the molecular pathways involved in hypoxic radiosensitization, and explore ways to target cancer stem cell populations with CIRT to improve outcomes for difficult-to-treat tumors.

## 5. Current Drawbacks of CIRT Technology

Despite the radiobiological advantage of high RBE, low OER, and the Bragg peak phenomenon, these in vitro properties need to be further explored in clinical settings. Carbon ions are more sensitive to range uncertainties, which can lead to either the “overkill effect” or subtherapeutic dosing, leading to suboptimal outcomes. This problem is compounded by the unavailability of CIRT-specific planning target volumes (PTV) and clinical target volumes (CTV) [3]. In a prospective stage II study of hypofractionated CIRT for peripheral non-small cell lung cancer, Saitoh et al. noted the 2- and 5-year survival rates to be around 90%, which was comparable to the study by Miyamoto et al., but noted higher percentages of local recurrences, highlighting the variability in treatment planning parameters [84,85]. While the RBE of proton beams may be assigned a value of 1.1 without being too far off from measured values for different tissues and different biological endpoints, the RBE values of carbon ions vary significantly over the SOBP, largely due to variations in linear energy transfer and energy. While LET can be averaged over fluence and dose along the SOBP, converting LET to RBE requires more sophisticated modeling, reviewed in great depth by others [9]. As with all RBE estimations, the specific cell/tissue type, its genetic makeup, and the biological endpoint chosen also influence the RBE [86]. For instance, CIRT inactivates tumor cells with p53 mutations or those overexpressing survivin or BCL-2 to a greater magnitude [9]. Different RBE values have been reported when CIRT is applied to different cell lines of the same cancer tissue type and also to diverse benign tissues (human dermal fibroblasts/HDF − 1.90 ± 0.0327; mammary ductal epithelium/hTERT-HME1 − 1.71 ± 0.0280; normal lung epithelium/NuLi-1 − 2.54 ± 0.0531) [87]. Efforts to generate a uniform SOBP across the tumor have required scaling factors for LET to RBE along the SOBP. Kim et al. have proposed a linear–quadratic model combined with Geant4-based Monte Carlo simulation code, where Bragg peak profiles and LETs were calculated for each slice in the target region, and weighting factors (power functions in terms of energy steps) were utilized to obtain a uniform SOBP [88]. In another strategy, a novel treatment planning system, which corrects for the enhanced biologic effect of carbon ions on cancer cells, is combined with Katz’s track structure model of cellular survival to more accurately predict the cell killing effect of primary carbon ions and the secondary fragments generated [89]. In patients with head and neck cancers, where dose escalation tends to enhance local control, innovative LET painting has been utilized to ensure uniform distribution of LET_d_ within the gross tumor volume while increasing the minimum LET_d_ [90]. First described by Bassler et al., LET painting prevents the dilution of average LET in the target volume to ensure delivery of maximal LET. This strategy can also potentially overcome tumor hypoxia in addition to allowing dose escalation [91].

The need for a synchrotron and additional shielding comes with structural and engineering complexities and high establishment costs, which in turn have led to a paucity of clinical centers with long-term experience with CIRT. This has led to data gaps and a lack of large-scale clinical trials. Takayasu et al. in their stage II trial on mucosal melanomas recorded that CIRT leads to good local control but with no clear difference in long-term control/overall survival among CIRT, protons, and X-ray therapy [92,93]. This is probably due to the competing risk of distant metastasis development (distant metastatic recurrence occurred in 92% of patients within 5 years and in 80% within 1 year), which is inadequately addressed by any localized treatment modality. The small number of participants in clinical trials involving CIRT is also a challenge and will need to be externally validated by other groups. Similarly, Kasuya G et al. noted excellent local control of hepatocellular carcinoma treatment using CIRT with minimal resultant hepatotoxicity but failed to record any significant improvement in the overall survival [94]. Scarce clinical data limit any head-to-head comparison of patient outcomes with the currently employed forms of RT, namely photon RT and proton RT. Systematic reviews and meta-analyses provide some insight, however. A retrospective analysis involving 53 patients treated with CIRT and 63 patients treated with photon RT for isolated paraaortic lymph node metastases after colorectal cancer resection noted that CIRT was associated with higher 2-year local control (78.9% vs. 68.1%), but no difference in overall survival or progression-free survival was noted. Notably, the CIRT group had no acute gastrointestinal toxicity but higher (22.6% vs. 17.5%) acute hematological toxicity. Delayed toxicity was absent in the CIRT group [95]. Recent results from choroidal melanoma cohorts treated with CIRT or photon RT are also encouraging. The CIRT cohort had a significantly superior 5-year progression-free survival (69% vs. 56.5%) and significantly reduced risk of local failure (5.6% vs. 13.4%), dramatically reducing the necessity for enucleation (8.5% vs. 24.2) [96]. Clinical trials comparing outcomes with proton RT are also sparse. A comprehensive systematic review comparing 947 (proton RT) and 910 (CIRT) patients treated for diverse cancer types reported a hazard ratio of 0.690 for local control after CIRT (*p* value = 0.031) with no statistically significant differences in progression-free or overall survival [97].

While extensive in vitro studies have explored the immunogenic potential of CIRT and its ability to inhibit distant metastasis, Ramie Shafi et al. recorded that though CIRT suppresses tumor migration in glioma cells, it conversely increases invasiveness in the Panc-1 cell line [10,98]. This raises concerns that the immunogenicity of CIRT is not homogeneous across malignant cell types and needs further in vitro analyses before it can be applied to varied tumor tissues.

Lastly, the dense ionization tracks and disintegration tail of CIRT raise concerns regarding the potential to induce subsequent primary cancers after a latent period, particularly in pediatric populations, if these inadvertently end up in normal tissues [2]. Despite this theoretical risk, however, patient cohort studies report that CIRT is associated with a lower risk of subsequent primary cancers compared to photon therapy (hazard ratio [HR] 0·81 [95% CI 0·66–0·99]; *p* = 0·038) in patients treated for prostatic carcinoma [99]. Further, a longitudinal follow-up of uterine cervical cancer patients also reports that while the difference in 10-year cumulative incidence is negligible (9.5% in CIRT and 9.4% in photon RT), the standardized incidence ratio (SIR) is modestly lower in CIRT-treated patients (1.1 in CIRT vs. 1.4 in photon RT) [100]. The available data, however, should be interpreted with caution as the CIRT cohorts tend to be smaller, and also, real-world clinical data for diverse cancer types are lacking.

## 6. Conclusions

CIRT is a form of high-LET radiation therapy that offers several theoretical advantages over traditional low-LET modalities. These include the ability to deliver high intratumoral radiation doses while largely sparing adjacent critical structures, the potential for robust immune activation that may lead to abscopal effects and limit metastatic spread, and the ability to radiosensitize typically radioresistant, hypoxic tumors. Given these unique characteristics, there are promising novel avenues to improve the therapeutic ratio of radiation treatment using CIRT, such as combining particle therapy with immunotherapy, delivering adaptive high-LET boosts to hypoxic regions, and safely delivering curative radiation doses to tumors located near critical organs. As interest in heavy ion therapy continues to grow, further preclinical and clinical studies, as well as randomized controlled trials, are needed to fully explore the immense potential of CIRT in improving global cancer outcomes.

## Figures and Tables

**Figure 1 cancers-18-00113-f001:**
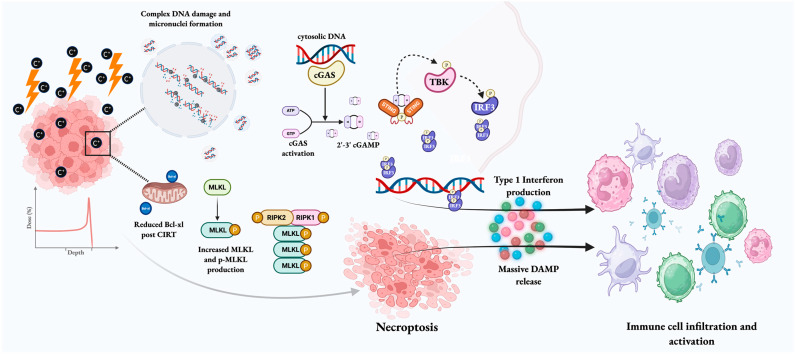
The immunogenic potential of CIRT. Given its high-LET nature, CIRT can cause extensive direct and irreparable DNA damage and the formation of cytoplasmic micronuclei. The cytoplasmic DNA is sensed by cGAS, leading to cGAMP synthesis, which then binds to STING. The STING complex translocates to the Golgi apparatus and recruits TBK1, which mediates the phosphorylation of IRF3, leading to transcription of Type I interferons. Type I interferons catalyze the maturation of dendritic cells and cause CD8+ T cell priming. Accumulation of phospho-MLKL because of Bcl-xl level decrease and complex unrepaired DNA damage triggers necroptosis. Dying tumor cells also release DAMPs that can mediate dendritic cell and immune cell recruitment to and activation at the tumor site. These actions ultimately facilitate the conversion of an immunologically “cold” tumor into a “hot” entity. [Created in BioRender. Biswal, P. (2025) https://BioRender.com/itrjcp6, accessed on 10 December 2025].

**Table 1 cancers-18-00113-t001:** Possible mechanisms of CIRT-induced radioimmunogenic cell death.

No.	Possible Mechanisms of Radioimmunogenic Cell Death	Cell Line/Disease Type	Ref.
1	↑ repulsive guidance molecule b (RGMb) and subsequent clonal expansion of tissue-resident memory T cells leading to pro-inflammatory cytokine production	Whole thoracic CIRT (C57BL/6J mice)	[45]
2	↑ IFN-γ pro-inflammatory cytokines and CD8+ T cell infiltration, and ↑ of ferroptosis gene signatures	Bilateral B16-OVA melanoma model (C57BL/6J mice)	[46]
3	cGAS-STING activation and ↑ TIL infiltration	Unilateral B16-F10 melanoma model (C57BL/6J mice)	[44]
4	↑ calreticulin, ATP, HMGB1, type-1 interferon and TILs in combination with anti-PD-1 therapy	Human U2OS osteosarcoma cells (in vitro) B16 and S91 melanoma models (C57BL/6J mice)	[37]
5	↓ expression of necroptosis inhibitor (caspase-8 and Bcl-x) leading to ↑ p-MLKL induced necroptosis and DAMP release	Nasopharyngeal cancer (CNE-2), hepatoma (HepG2), and laryngeal cancer (Hep2) cells (in vitro)	[27]
6	↑ pro-inflammatory cytokines (granzyme B, IL-2, and TNF-α, IFN-γ, IL-2, and IL-1 beta) and CD8+ T cells No proliferation of immunosuppressive Treg with low-dose CIRT, unlike photon RT	Orthotopic 4T1 model (C57BL/6J mice)	[47]
7	↑ HMGB1 secretion while reducing anti-inflammatory cytokines (IL-10 and TGF-β)	A549, H520, and Lewis Lung cancer cells (in vitro)	[48]
8	↑ PDL-1 greater than X-Ray RT via STAT1-ATR-IRF1 pathway	Human U2OS osteosarcoma cells (in vitro)	[41]
9	↑ KLRK1 gene expression and NKG2D/NKG2D-Ls pathway activation ↑ NK cell function and infiltration	Lewis lung cancer xenograft (C57BL/6J mice)	[49]
10	↓ MDSC population in spleen, blood, and bone marrow after CIRT via JAK2/STAT3-dependent mechanism, resulting in ↑ TILs and macrophages	B16, MelanA and S91 melanoma xenografts (C57BL/6J mice)	[50]
11	Delayed (72 h) ↑ of STING-STAT1 axis (HMGB2, OAS1, HLA-B)	Human KYSE450 esophageal carcinoma (in vitro)	[51]
12	↑ TNF-α signaling via the NF-κB pathway and alternative splicing of HLA class I and II molecules and histone-coding genes	Peripheral blood from human donors after CIRT and human lymphoblastoid cells (TK6, WTK1, and NH32) (in vitro)	[52]
13	↑ FLT3LG transcription	Liquid biopsy samples from recurrent CIRT-treated high-grade glioma patients	[53]
14	↑ proliferation of lymphocyte proliferation ↑ TNF secretion	Peripheral blood samples from CIRT-treated localized prostate cancer patients	[54]
15	↑ NK cell activation, ↑ TNF and IL1 genes ↑ naïve T cell activation in the abscopal unirradiated tumor	Murine Her2+ EO771 xenografts (C57BL/6J mice) treated with CIRT and anti-CTLA4	[55]
16	↑ surface calreticulin expression at low grade (2Gy and 4Gy) CIRT compared to photon and proton-RT	Human A549 (lung adenocarcinoma), U251MG (glioma), Tca8113 (tongue squamous carcinoma), and CNE-2 (nasopharyngeal carcinoma) cells (in vitro)	[33]
17	↓ infiltration of microglia and myeloid-derived suppressor cells, thereby blocking the HIF1-α /stromal cell-derived factor 1/CXCR4 axis ↑ CD8+ T cell infiltration and reduced M2 macrophage polarization	Murine radioresistant NCH644, NCH441, and T3259 (glioblastoma stem cells) orthotopic xenografts (C57BL/6J mice)	[56]
18	Dose and time dependent ↑ in HMGB1 secretion after CIRT	Human KYSE70 (esophageal carcinoma), HeLa, and SiHa (cervical cancer) cells (in vitro)	[57]
19	↑ CD3+ tumor infiltration 24–48 hr after CIRT compared to equivalent RBE X-Ray RT	Murine Soft tissue sarcoma model (Kras^LSL-G12D^; p53^fl/fl^KP mice)	[58]
20	↑ expression of NF-κB targets, TNF, and CXCL genes	Human HEK-pNF-kappa B-d2EGFP/Neo L2 cells (in vitro)	[59]
21	↑ IFN-γ and IL-12 in LPS or CpG-treated DCs after CIRT	GM-CSF and IL-4 co-cultured bone marrow cells to induce dendritic cell differentiation (in vitro)	[60]
22	Dendritic cell infusion after CIRT reduced pulmonary metastases ↑ DC maturation (↑ CD40 and IL-12) on co-culture with CIRT-treated cells	Murine NR-S1 (oral cancer) bearing xenograft (C3H/He mice) and NR-S1 cells (in vitro)	[36]
23	↑ HMGB1 levels in cell supernatant 72 and 96 hr after CIRT	Human TE2 (esophageal cancer), KYSE70 (esophageal cancer), A549 (lung adenocarcinoma), NCI-H460 (large cell lung cancer), and WiDr (colorectal cancer) cells (in vitro)	[61]
25	↓ of unirradiated pathologic para-aortic lymph nodes—possible abscopal immune response	Patient with underlying ATM exon 46 frameshift mutation in clear cell adenocarcinoma of the cervix =	[62]
26	DNA damage repair-deficient (BRCA1 mutation) cells strongly ↑ the NF-κB-driven immunogenic response after CIRT CIRT ↑ IRF3, IFNB1 and pSTAT1 more potently than X-Ray RT Autophagy inhibition ↑ cGAS-STING and NF-κB activation	Human MDA-MB-436, MDA-MB-468, and MDA-MB-231 TNBC cells (in vitro)	[63]
27	↑ cytoplasmic dsDNA and cGAS-STING compared to photon RT	Murine 4T1 cells (in vitro)	[64]
28	↓ Tumor growth and ↑ CD8+ T cell infiltration in CIRT + hydroxychloroquine-treated tumors	Pan02 xenografts inoculated with cells treated with CIRT alone, autophagy inhibitor (HCQ) alone or combination (C57BL/6J mice)	[65]
29	CIRT ↑ calreticulin and PD-L1 on B16F10 and Pan02 cell line, while ultra-high dose rate CIRT ↓ PD-L1 on Pan02	Murine B16F10 (melanoma) and Pan02 (pancreatic adenocarcinoma) xenografts treated with conventional and ultra-high dose rate CIRT	[66]
30	ATR and ATM inhibitors ↑ IFN-3 secretion and STAT-1 activation synergistically with CIRT	Human glioblastoma (U-251 and T98G) cells, in combination with ATR and ATM inhibitors (in vitro)	[67]
31	↑ CD8+ T cell infiltration and IFN-β secretion, more so with higher doses ↑ PD-L1 and TREX-1 after CIRT	Bilateral Lewis lung carcinoma xenograft (C57BL/6J mice)	[68]
32	cGAS-STING activation and ↑ production of downstream cytokines and chemokines (CCL5, CXCL10, and IFNβ1) ↑ tumor macrophages, splenic CD8+ T cells and CD8+ effector memory T cells ↑ IFN-γ production by CD8+ TILs	Murine RM1 xenografts (C57BL/6J mice and BALB/c nude mice)	[69]

Abbreviations: ATP—adenosine triphosphate; ATM—ataxia telangiectasia mutated; ATR—ataxia telangiectasia and Rad3-related; BRCA1—breast cancer gene 1; cGAS-STING—cytosolic GMP-AMP synthase—stimulator of interferon genes; CCL—C-C motif chemokine ligand; CIRT—carbon ion radiation therapy; CpG—cytosine linked to guanine via a phosphate; CXCL—C-X-C motif chemokine ligand; CXCR4—C-X-C motif chemokine receptor type 4; DAMP—damage-associated molecular pattern; DC—dendritic cells; FLT3LG—Fms-related tyrosine kinase 3 ligand; HLA-B—human leucocyte antigen-B (major histocompatibility complex class I B; HMGB1—high mobility group box 1; IFN—interferon; IL—interleukin; IRF—interferon regulatory factor; JAK2—Janus kinase 2; KLRK1—killer cell lectin-like receptor K1; LPS—lipopolysaccharide; MLKL—mixed lineage kinase domain like; NF-κB—nuclear factor—kappa light-chain-enhancer of activated B cells; NKG2D—natural killer group 2, member D; NKG2D-L—NKG2D ligand; OAS1—2′-5′ oligoadenylate synthase 1; PD-1—programmed cell death protein 1; PDL-1—programmed cell death ligand-1; RBE—relative biological effectiveness; RGMb—repulsive guidance molecule b; STAT—signal transducers and activators of transcription; TGF—tumor growth factor; TILs—tumor infiltrating lymphocytes; TNF—tumor necrosis factor; Treg—regulatory T cell; TREX-1—three prime repair exonuclease 1; ↑—increase; ↓—decrease.

**Table 2 cancers-18-00113-t002:** Status of clinical trials involving CIRT and immunotherapy.

NCT ID	Title	Phase	Disease	Agent	Status
NCT02946138	Phase II trial of CIRT combined with GM-CSF for the treatment of hepatocellular carcinoma	II	Hepatocellular carcinoma	Concurrent GM-CSF	Withdrawn
NCT07246668	A Phase II single-arm clinical study to evaluate the efficacy and safety of CIRT with atezolizumab and bevacizumab combination therapy in patients with advanced hepatocellular carcinoma	II	Hepatocellular carcinoma	Atezolizumab (anti-PD-L1) maintenance therapy after CIRT	Recruiting
NCT05478876	CIRT in the treatment of mucous melanomas of the female lower genital tract (CYCLE)	NA	Pelvic recurrent disease of gynecological cancers	Not specified; immunotherapy after CIRT	Recruiting
NCT07035860	Efficacy and safety of chemoimmunotherapy and CIRT in unresectable locally advanced non-small cell lung cancer	Observational	Non-small cell lung cancer	Anti-PD-L1 (Durvalumab, Sugemalimab, Atezolizumab, Benmelstobart) or Anti-PD-1 (pembrolizumab, camrelizumab, toripalimab, tislelizumab, sintilimab, nivolumab, serplulimab, penplulimab) neoadjuvant and adjuvant to CIRT	Not yet recruiting
NCT01245985	TPF followed by cetuximab and IMRT plus carbon ion boost for locally advanced head and neck tumors (TPF-C-HIT)	II	Locally advanced squamous cell carcinoma of the head and neck	Cetuximab and carbon ion boost after TPF chemotherapy	Terminated
NCT04143984	CIRT plus camrelizumab for locally recurrent nasopharyngeal carcinoma	II	Locally recurrent nasopharyngeal carcinoma	Anti-PD-1 (camrelizumab) after CIRT	Recruiting
NCT05229614	Immunotherapy and CIRT in solid cancers with stable disease (ICONIC)	II	Non-small cell lung cancer, head and neck squamous cell carcinoma, melanoma, and urothelial carcinoma	Anti-PD-1 (pembrolizumab) concurrent with CIRT	Recruiting
NCT06311981	CIRT for locally advanced lung cancer in elderly patients	II	Locally advanced non-small cell lung cancer	Anti-PD-1 (pembrolizumab) after CIRT	Recruiting
NCT06805864	Pembrolizumab concurrent with and following carbon-ion radiotherapy for locally advanced cervical adenocarcinoma (BROTHER)	II	Adeno and adenosquamous carcinoma cervical carcinoma	Concurrent and adjuvant anti-PD-1 (pembrolizumab)	Not yet recruiting
NCT01192087	Adenoid cystic carcinoma, erbitux, and particle therapy (ACCEPT)	I/II	Adenoid-cystic carcinoma of the head and neck	Neoadjuvant and concurrent cetuximab	Unknown

Abbreviations: CIRT—carbon ion radiation therapy; GM-CSF—granulocyte–monocyte–colony-stimulating factor; IMRT—intensity modulated radiation therapy; PD-1—programmed cell death protein 1; PD-L1—programmed cell death ligand-1; TPF—chemotherapy combination of taxotere (docetaxel), platinum (cisplatin), and 5-fluorouracil.

## Data Availability

Not applicable.

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
