# Peer review of "Novel Physical and Biological Applications of Carbon Ion Radiotherapy"

_cancers, 2025, doi:10.3390/cancers18010113_

Round 1
Reviewer 1 Report
Comments and Suggestions for Authors
This manuscript provides a well-written and comprehensive review of the unique physical and biological properties of Carbon Ion Radiotherapy (CIRT), with a focus on its novel applications. The discussion on immunomodulation and overcoming hypoxia is timely and relevant. The paper is generally well-structured and clearly presented. However, several sections would benefit from revision to improve clarity, specificity, and the overall impact of the review. The following major and minor points are offered to help strengthen the manuscript.
Major comments
Figure 1: The mechanism depicted in Figure 1 (cGAS-STING pathway activation) is a general mechanism for radiation-induced immunity and is not specific to CIRT. Both X-rays and CIRT can induce this pathway. To effectively highlight the unique immunogenic potential of CIRT, as intended by the manuscript, this figure should be modified. We suggest illustrating the key differences in immune activation between low-LET (e.g., X-ray) and high-LET (CIRT) radiation, such as the differential induction of complex DNA damage, higher rates of micronuclei formation, or the activation of alternative cell death pathways (like necroptosis) which are discussed in the text.
LINE 219-230 (Discussion on ICI combination): The theoretical background for combining CIRT with immune checkpoint inhibitors (ICIs) is a recent "hot topic." This section could be a stronger focal point of the review. We recommend expanding this discussion. For instance, to add depth, consider including the mechanistic link between DNA double-strand breaks (a hallmark of high-LET radiation) and PD-L1 expression, as reported by Sato et al. (https://doi.org/10.1038/s41467-017-01883-9).
LINE 256-258 (Abscopal effects): The authors state that "further studies are necessary." While true, this statement would be strengthened by acknowledging that several clinical studies combining particle therapy (including CIRT) with ICIs are already ongoing. Citing a few of these registered trials or protocols would provide a more current and forward-looking perspective on the "therapeutic potential of this concept."
LINE 281-286: The citation of Kasuya et al. (ref 46) discusses clinical outcomes for hepatocellular carcinoma (HCC) and is not directly related to the topic of tumor hypoxia. Furthermore, comparing these HCC outcomes to the proton therapy studies (refs 47, 48) is challenging due to different patient populations and disease contexts. This citation feels misplaced within the hypoxia/RBE discussion. The authors should reconsider this reference. A citation to a basic or translational study that more directly demonstrates the efficacy of CIRT in hypoxic models compared to low-LET radiation would be more appropriate here.
LINE 332-364 (“Current Drawback of CIRT Technology” section): A significant drawback and critical challenge for CIRT that is not adequately discussed is the biological uncertainty, specifically the heterogeneity of the Relative Biological Effectiveness (RBE) within the SOBP and its dependency on dose/fractionation, tissue type, and endpoint. This is a major issue to be overcome for its wider application. The review would be improved by including this point and mentioning emerging technologies designed to address it, such as "LET painting" or other biological dose-painting strategies.
LINE 363 (Secondary cancers): When discussing the potential for secondary cancers, the review would benefit from citing and discussing the significant findings from large-scale clinical follow-up studies, such as the analysis from the NIRS/QST group in Japan (e.g., DOI: 10.1016/S1470-2045(18)30931-8). This would provide important clinical context to the discussion of long-term toxicity.
Minor comments
LINE 200-205 (Citation check): Please double-check the citations for references [35-37]. The content attributed to these references in the text (murine models, combination immunotherapy) seems to differ from the actual content of the cited papers. For example, Ref [36] appears to be a Phase I trial protocol for HCC, not a murine model study. Please verify and correct these citations.
LINE 208 (Missing citation): The sentence begins, “Studies to date have shown that…”. Please provide the specific citation(s) to support this claim.
Comments on the Quality of English LanguageThe quality of the English language is generally good, and the manuscript is clearly understandable.
Reviewer 2 Report
Comments and Suggestions for Authors
In this review paper, the authors summarize the physical and radiobiological advantages of carbon ion radiotherapy (CIRT), its potential role as an immunogenic cancer therapy, and the disadvantages of this radiotherapy modality. The review is comprehensive and up to date. However, I believe the article would have been more informative if a subtitle, and perhaps a table, had been added describing clinical studies in which CIRT appears superior to proton- or photon-based radiotherapy.
Minor comment:
Introduction: Authors should state the value or approximate value of the RBE and discuss the problems associated with its determination.
Reviewer 3 Report
Comments and Suggestions for Authors
The article "Novel Physical and Biological Applications of Carbon Ion Radiotherapy" is a comprehensive and well-structured review of the potential of CIRT, which analyzes the physical aspects, biological mechanisms, and emerging clinical applications.
The difference between low LET radiation and carbon ions is clearly explained, and the poor lateral diffusion of the beam and the lower dependence on oxygen are also highlighted, with clear links to tumor pathophysiology.
In my opinion, the analysis of CIRT's ability to activate the immune system more than photons and the potential transformation of a "cold" tumor into a "hot" one is very interesting.
The preliminary data on the possible abscopal effect is also interesting, and the authors' prudent position is appreciated, as they underline the need for further research.
The section on overcoming radioresistance from hypoxia is well supported by preclinical studies and clinical data on malignancies that are difficult to treat and have a poor prognosis.
The review clearly addresses the limits of CIRT: very high costs, beam range uncertainties, absence of randomized trials and non-uniform biological results, avoiding an overly optimistic view.
In my opinion, the article is well written and scientifically sound, comprehensive both from a radiobiological and clinical point of view, up-to-date and balanced in describing limits and potential.
I believe it can be published in its present form.
Reviewer 4 Report
Comments and Suggestions for Authors
Dear authors,
please find my comments in the attached file.
Best regards
